# Protein-peptide association kinetics beyond the seconds timescale from atomistic simulations

Fabian Paul[1,2], Christoph Wehmeyer [1], Esam T. Abualrous[1], Hao Wu[1], Michael D. Crabtree [3], Johannes Schöneberg[1], Jane Clarke[3], Christian Freund[4], Thomas R. Weikl[2] & Frank Noé [1]

Understanding and control of structures and rates involved in protein ligand binding are essential for drug design. Unfortunately, atomistic molecular dynamics (MD) simulations cannot directly sample the excessively long residence and rearrangement times of tightly binding complexes. Here we exploit the recently developed multi-ensemble Markov model framework to compute full protein-peptide kinetics of the oncoprotein fragment $^{25\text{-}109}$Mdm2 and the nano-molar inhibitor peptide PMI. Using this system, we report, for the first time, direct estimates of kinetics beyond the seconds timescale using simulations of an all-atom MD model, with high accuracy and precision. These results only require explicit simulations on the sub-milliseconds timescale and are tested against existing mutagenesis data and our own experimental measurements of the dissociation and association rates. The full kinetic model reveals an overall downhill but rugged binding funnel with multiple pathways. The overall strong binding arises from a variety of conformations with different hydrophobic contact surfaces that interconvert on the milliseconds timescale.

[1] Department of Molecular and Cell Biology and California Institute for Quantitative Biosciences, University of California, Berkeley, CA 94720, USA. [2] Max Planck Institute of Colloids and Interfaces, Department of Theory and Bio-Systems, 14476 Potsdam, Germany. [3] Department of Chemistry, University of Cambridge, Cambridge CB2 1EW, UK. [4] Institute of Chemistry and Biochemistry, Freie Universität Berlin, Thielallee 63, 14195 Berlin, Germany. Christoph Wehmeyer and Esam T. Abualrous contributed equally to this work. Correspondence and requests for materials should be addressed to F.N. (email: frank.noe@fu-berlin.de)

In the past, drug design has primarily focused on finding inhibitors with maximal binding affinity to the target. Recently, there has been a growing interest in optimizing target-drug kinetics[1, 2]. A direct strategy to exploit kinetics is the maximization of the drug's residence time at the receptor in order to ensure contiguous drug effect between subsequent deliveries[3, 4]. Protein–ligand kinetics may involve more than two kinetically relevant states, either due to different ligand binding poses, different protein conformations or their coupling[5–10]. While this multi-state nature is not always apparent in ensemble kinetic experiments[11], accounting for it may help during multiple stages of the drug design process[12, 13]. On the molecular scale, targeting receptor binding pockets that open transiently can lead to allosteric inhibitors[14, 15]. On the pharmacokinetic scale, a complete assessment of protein–drug kinetics can provide more accurate models and offer additional freedom to optimize the drug delivery strategy[2, 16]. Multi-state kinetics are especially relevant in multivalent binders, which are characterized by highly non-exponential kinetics and nonlinear amplification of the binding strength through multiple parallel binding interfaces[17, 18].

Simultaneous study of molecular structure and kinetics at high resolution is possible with fully flexible all-atom molecular dynamics (MD) simulation in explicit solvent. However, such simulations are limited to lengths of few microseconds on publicly available hardware. Few milliseconds can be reached on specialized hardware[19] or in aggregate times using distributed computing[20–23]. These simulation times are short compared to residence times of most high-affinity binders.

Calculating unbiased long-term kinetics for all-atom MD models is one of the hardest problems in molecular simulation, as it depends upon the solution of three difficult tasks simultaneously: (A) the ability to explore initially unknown states and conformational changes, (B) the repeated sampling of the slowest transitions, (C) the computation of unbiased transition rates from such simulation data. Fortunately, tools have been established that each excel at one or two of these tasks, and that can be combined to a powerful framework.

Path sampling and milestoning-based methods[24–27] enhance the probability of transition pathways between a priori known end-states and can be extended to compute transition rates (tasks B, C), but offer only limited help in exploring the state space. In contrast, unbiased MD simulations, especially high-throughput MD simulations[28, 29] can explore the state space without hindrance from constraints (task A). When analyzed with kinetic models, such as Markov state models (MSMs)[30–33], the unbiased long-term kinetics can be approximated[34, 35], without required initial knowledge of relevant states, coordinates or a timescale separation (task C). However, this approach relies on having sampled the rare-event transitions in the data. While MSMs help with parallelizing this problem and rare events can be sampled, in particular when adaptive sampling strategies are combined with high-throughput simulation[23], the sampling of very rare events such as protein-inhibitor dissociation can still be very inefficient. In practice, this difficulty may result in not properly connected models and underestimated or imprecisely estimated residence times. While MSM analyses have the advantage of being able to detect these problems with carefully conducted Markovianity tests[36] and by computing binding free energies as a function of the MSM lag time[37, 38], the typical solution involves running more simulations, which is unpractical when computational resources are limited. Enhanced sampling methods such as umbrella sampling, flooding, metadynamics, or replica exchange[39–42] are specialized in rare-event sampling (task B), and some of them can significantly help to explore states with low populations (task A), however they rely on a priori knowledge of good collective coordinates. Kinetic quantities cannot be directly computed from such data and the data analysis relies on the applicability of macroscopic rate theories[43]. This has been mitigated by recent progress in hyper-dynamics which allows to predict transition rates between long-lived states when good collective coordinates are known[44–48].

In order to combine the advantages of enhanced sampling methods and MSMs, we recently developed the concept of multi-ensemble Markov models (MEMMs)[49]. MEMMs rely on the idea of combining unbiased simulations of fast events (such as rapid binding) with efficient sampling of the rare events in biased ensembles (such as biased unbinding) within a reweighting framework that can extract full and unbiased kinetics. Several MEMM estimators have been developed[50–52], including the statistically optimal transition-based reweighting analysis method (TRAM), which exploits detailed balance to extract unbiased kinetics of the slow steps from equilibrium properties harvested at biased ensembles[49, 53]. The recently introduced bin-less TRAM version can compute complex multi-state kinetics without requiring pre-defined collective variables[49], which allows kinetics in very high-dimensional and complex examples to be studied.

Here we show how enhanced MD simulation techniques can be combined to compute unbiased multi-state kinetics of the onco-protein fragment $^{25–109}$Mdm2 with the nano-molar peptide inhibitor PMI in all-atom resolution. MEMMs are the key technology for this achievement, and allow us to obtain the residence time that is beyond the seconds timescale with high accuracy and precision, from sub-millisecond simulations. Multiple intermediates and mis-bound modes are found, the equilibrium folding–binding pathways are computed. The simulations are tested against previous mutagenesis experiments and binding–unbinding kinetics experiments conducted here.

## Results

**Direct MD simulation of protein–ligand complex Mdm2–PMI.** Mdm2 is a major therapeutic target that antagonizes the tumor suppressor p53 by ubiquitinating it or by binding the N-terminal trans-activation domain (TAD) of p53. In certain cancers, Mdm2 is over-expressed leading to excessive inactivation of p53[54]. Therefore the Mdm2–p53 interaction is a primary target for inhibitor design[55–57]. The 12-amino-acid peptide PMI (p53–Mdm2/MdmX inhibitor) is one of the strongest known Mdm2 binders, with a dissociation constant of $K_d = 3.3$ nM[57]. In the co-crystal structure of PMI with the protein fragment $^{25–109}$Mdm2, PMI binds as a helix[57] while our MD simulations of PMI without its binding partner suggest that PMI is at most 40% helical in isolation. Thus the binding mechanism must involve PMI folding. The binding of PMI to the Mdm2 protein fragment is a particular challenging system for MD not only because of the high affinity but also because of the abundance of metastable states that act as traps on achievable simulation lengths of microseconds. In Zwier et al.[58], 120 μs of implicit solvent simulations of the same Mdm2 fragment were conducted with a different p53-peptide and only 10% of the simulations reached the crystallographic binding pose.

We conducted 500 μs of unbiased atomistic MD simulations of the protein fragment $^{25–109}$Mdm2 and the PMI peptide from different initial structures, especially dissociated states. A preliminary analysis showed that these trajectories contain five complete binding events from dissociated to crystal-like states, several tens of partial binding events via intermediates. A variety of intermediates and trap states were found (Fig. 1). However, not a single clear dissociation event was observed, and a MSM constructed from the unbiased MD data contained many disconnected states.

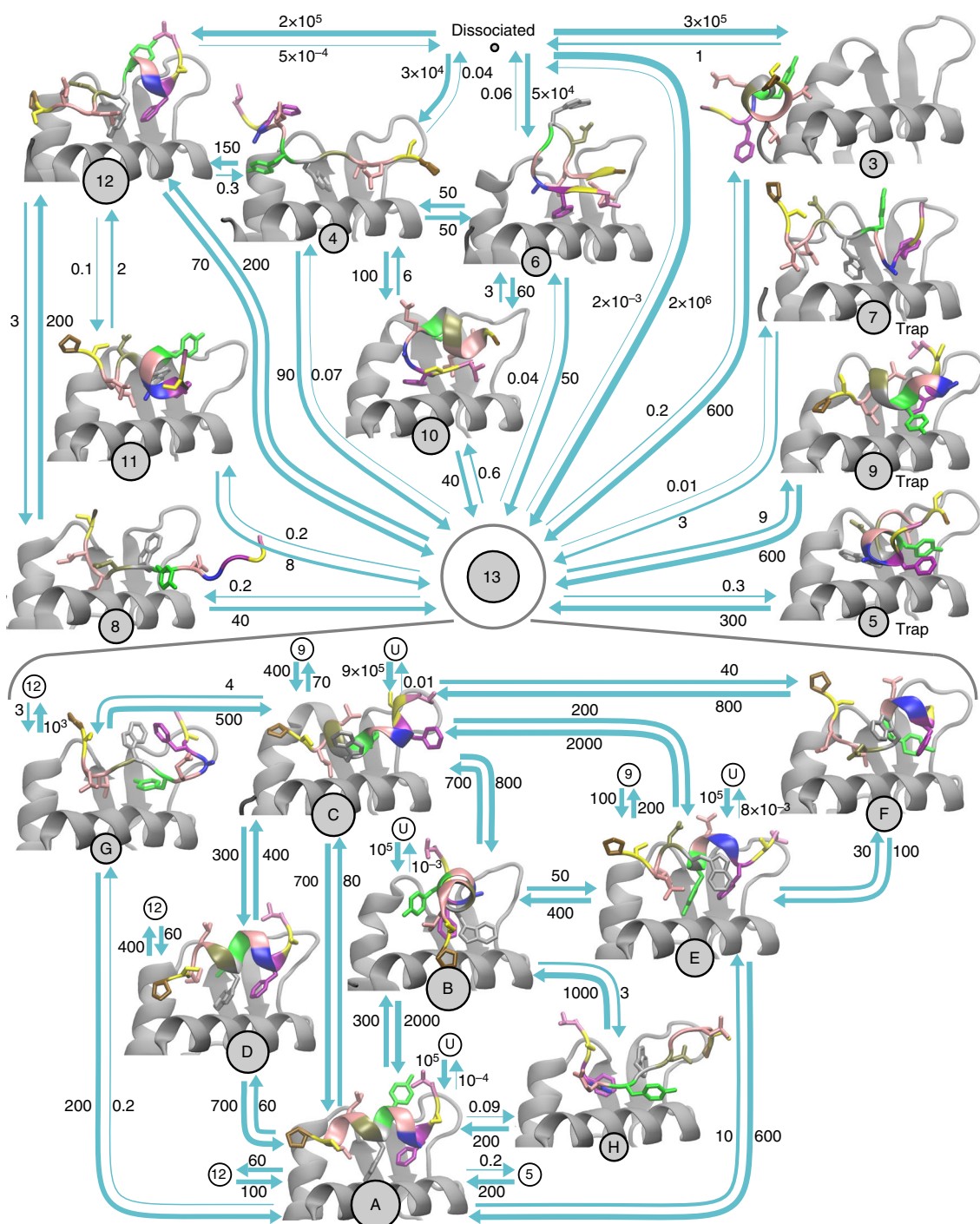

**Fig. 1** Metastable states and transition rates for the binding of PMI to Mdm2. The PMI peptide is colored according to (TSFAEYWNLLSP). States are represented by discs with areas proportional to the natural logarithm of the equilibrium probability. Arrows indicate transitions with rate constants of at least 1 ms$^{-1}$ in either direction. Numbers quantify transition rate constants in ms$^{-1}$ M$^{-1}$ for association events and in ms$^{-1}$ for all other transitions. The definition of the states is hierarchical: between top-level states 0 and 13, transitions happen on timescales of 10 μs or slower. States in the lower part of the figure are sub-states of top-level state 13. There, PMI transitions between different states in the main binding pocket of Mdm2 on timescales of microseconds or slower (only states with large probabilities are shown)

**Biased simulations predict the binding affinity.** Consequently, we added biased simulations with the aim of reversibly sampling bound, unbound and intermediate states. MEMMs can in principle be built using any biased sampling protocol, including umbrella sampling[39] or metadynamics[41]. Here, six independent Hamiltonian replica-exchange simulations were conducted, each about 1 μs long and with 14 replicas. The first Hamiltonian is unbiased while the other Hamiltonians have gradually reduced protein–ligand interaction strengths (see "Methods"). In contrast to unbiased MD, these simulations do not provide direct kinetic information, but sampled efficiently different binding sites and binding modes. After discarding the initial equilibration phase of 50 ns (Supplementary Note 3.3) these data still contained six full binding and 26 full

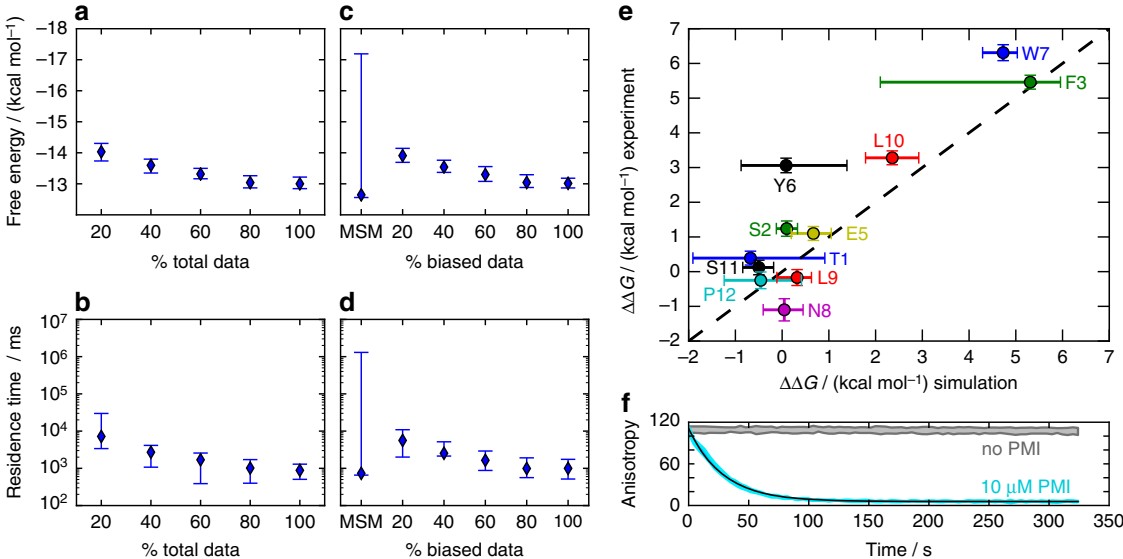

**Fig. 2** Computational predictions and experimental validations of binding affinities and kinetics of PMI–Mdm2. **a, b** Maximum likelihood estimates of the binding free energy and residence time, respectively, as a function of the amount of data used. Diamonds mark maximum likelihood estimates, error bars indicate 95% confidence intervals. *x*% of total data means that *x*% of all biased data and *x*% of all unbiased data were used. **c, d** Estimates of binding free energy and residence time as a function of data composition. The fraction of biased simulation data is varied between 0 and 100% of all biased data while keeping the sum of the amount of biased and the amount of unbiased data constant at 502 μs. **e** Validation of the simulation model: binding free energy differences ($\Delta\Delta G$) of PMI–Mdm2 upon alanine mutations of the PMI peptide, compared between the present simulations and experiments[59]. Only biased simulation data was used and analyzed with MBAR[62]. Error bars mark standard deviations (simulation error computed using bootstrap, see Supplementary Note 3.3). **f** Cyan: average and standard deviations of anisotropy time traces from three repeats of a binding competition experiment, Mdm2 (10 nM), pre-incubated with labeled PMI (10 nM), was mixed with unlabeled PMI (10 μM). A mono-exponential function (black) was fitted to the average time trace. Gray: control without unlabeled PMI

dissociation events, as well as many transitions between intermediates (Fig. 1).

To test the enhanced sampling simulation, we determined the dissociation constant between Mdm2 and PMI experimentally using fluorescence anisotropy (see Supplementary Note 4.2 and Supplementary Fig. 11), obtaining $K_d^{\mathrm{exp}} = 3.02 \pm 0.31$ nM, in agreement with previous data[59]. Computationally, $K_d^{\mathrm{sim}} = 0.34$ nM (95% confidence interval: [0.22 nM, 0.44 nM]) was determined by applying the PyEMMA implementation[60] of the MBAR estimator[61, 62] on the replica-exchange data (Supplementary Note 3.6). The difference between the computational and the experimental value corresponds to 1.3 kcal mol$^{-1}$, which is in the expected range of force field inaccuracies[63, 64]. As a more comprehensive test, previously measured changes in binding free energies ($\Delta\Delta G$) upon mutation of PMI residues to alanine were predicted using perturbation theory[65, 66]. We find good agreement of the $\Delta\Delta G$ values between simulation and experiment[59] within statistical uncertainties, in particular for the amino acids that are important for binding: Phe3, Trp7, and Leu10 (Fig. 2e and Supplementary Note 3.2).

**Multi-ensemble Markov models reveal slow unbinding kinetics**. We developed an extension of the recent TRAM estimator[49] called TRAMMBAR for combining unbiased MD simulations with replica-exchange simulations (see "Methods"). While TRAM requires all simulations to be longer than its lag time (often on the order of tens to hundreds of nanoseconds), this is not the case for replica-exchange simulations with rapid exchanges. TRAMMBAR can employ such replica-exchange data, by assuming global equilibrium for that part of the simulation, which is justified when statistical tests indicate short correlation times[62]. The present replica-exchange data has a correlation time of 40 ns,

compared to simulation lengths of about 1 μs (Supplementary Note 3.3). Using TRAMMBAR, all unbiased and biased simulation data were combined to a MEMM with 1056 states at a lag time of 150 ns, and its self-consistency was validated using standard tests[35] (Supplementary Note 3.4 and Supplementary Figs. 5 and 6). The kinetics of the unbiased ensemble was then analyzed.

The association rate is predicted to be $3.3 \times 10^9$ M$^{-1}$ s$^{-1}$ (see "Methods", Supplementary Note 3.7) which is faster than the association of similar p53-peptides to the full-length N-terminal domain of Mdm2 (on the order of $10^7$ M$^{-1}$ s$^{-1}$)[67] and still faster than the association of the $^{17-29}$p53 peptide to the $^{25-109}$Mdm2 fragment ($k_{\mathrm{on}} = 7 \times 10^7$ M$^{-1}$ s$^{-1}$)[58]. The majority of association trajectories enter basin 13 that contains the crystallographic complex and is correctly predicted as the most populous state (Fig. 1).

Computing the residence time of the complex from the transition matrix may lead to a systematic overestimate, because the dissociated state lifetime is shorter than the lag time used to estimate the transition matrix. To avoid this bias, we estimated rate matrices. Rate matrix estimation is not unique and we considered the maximum likelihood approach of Kalbfleisch and Lawless[68] which gives an estimate of the residence time of 0.88 s (95% confidence interval [0.48 s, 1.33 s], see Fig. 2b, d), and the least-squares approach of Crommelin and Vanden-Eijnden[69], which gives an estimate of 8 s (confidence interval [1.5 s, 40 s]). To test the predicted values from the simulations, we decided to measure the binding kinetics of PMI to Mdm2 experimentally. We performed a binding competition experiment with a fluorescence anisotropy readout to measure the PMI dissociation rate and stopped-flow kinetics experiments to measure the association rate (see Supplementary Notes 4.1–4.3, Fig. 2f and Supplementary Fig. 12). We measured a residence time of

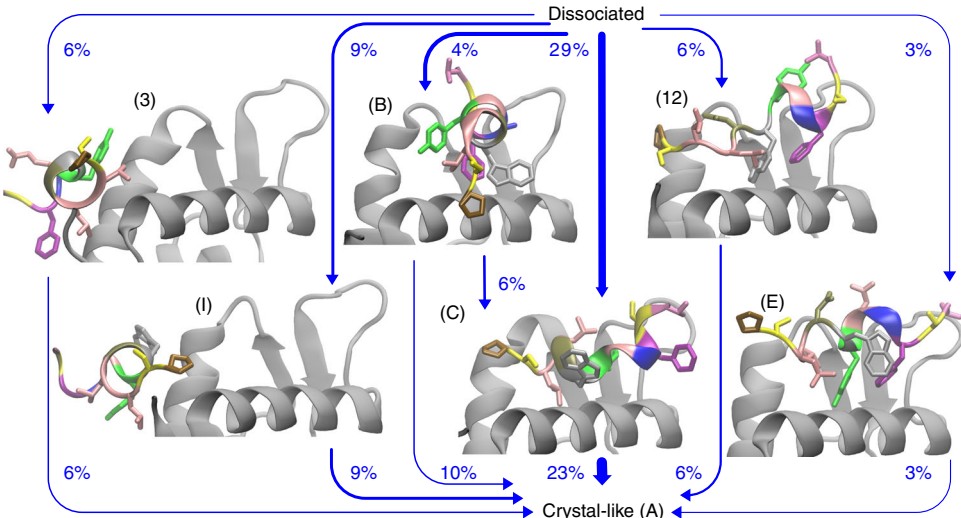

**Fig. 3** Binding mechanism comprised by the 60% most probable pathways. Structures of metastable (on-pathway) intermediates are shown, labels are as in Fig. 1. Arrows indicate the direction and relative magnitude of the reactive flux from the dissociated state to the crystal-like bound state. PMI residues that form PMI–Mdm2 contacts with at least a probability of 0.5 in a given macro-state are shown as sticks

26.8 s (confidence interval [24.7 s, 34.1 s]) and an association rate constant of $5.27 \times 10^8\,\mathrm{M}^{-1}\,\mathrm{s}^{-1}$ (confidence interval $[5.17, 5.37] \times 10^8\,\mathrm{M}^{-1}\,\mathrm{s}^{-1}$). Interestingly, our simulation-based predictions and the experimental estimate for the residence time all lie in the range of seconds to tens of seconds, which is a good agreement considering expected errors in the simulation force field[63, 64] and influence of the measurement by the fluorescence label. About 50% of the simulation data, i.e., a total of 300 μs of mixed unbiased and biased data, are sufficient to get estimates that are statistically indistinguishable from the estimates using all data (Fig. 2a, b and Supplementary Fig. 9a).

To assess the importance of the biased simulations for the computation of the binding free energy and the residence time, we varied the fraction of biased data used for the estimation (Fig. 2c, d and Supplementary Fig. 9b). Both quantities converge within statistical uncertainty if at least 50% of the biased data is included in the estimation (i.e., 450 μs unbiased data and a total of 50 μs biased data in all replicas). If no biased simulation data is used and a conventional MSM is estimated (using 500 μs unbiased data) the errors increase by a magnitude that makes the estimate practically useless. Note that it is not easy to determine whether a MSM is truly connected, and it is possible that this large error actually indicates that the dissociation pathway has not been sampled in the unbiased simulations alone.

**Analysis of the full kinetic network**. To obtain an overview of structure-kinetics relationships, we analyzed the MEMM kinetics between the dissociated state (protein-peptide distances larger than 1 nm) and 14 metastable states that interchange on the timescale of 10 μs or slower (Fig. 1 upper half). At this resolution, the binding is overall downhill with fast direct association rates on the order of $10^9\,\mathrm{M}^{-1}\,\mathrm{s}^{-1}$ into the native basin 13 that dominate the experimentally measurable on-rate. Association can also occur to non-native intermediates (3, 4, 6, 12) with smaller rates of $10^7$ to $10^8\,\mathrm{M}^{-1}\,\mathrm{s}^{-1}$ (Fig. 1).

In the most populous state 13, PMI is folded and anchored, with a high probability, to the binding pocket with its hydrophobic residues Phe3 and Trp7. In the second-most populous state 12, PMI has the folded crystallographic N-terminal conformation, but the C-terminus is unfolded and forms a different contact pattern: while Leu9 forms multiple contacts

with Mdm2 helix 2 (Supplementary Fig. 4), Leu10 has no contact to Leu54, Val93, and Ile99. Ser11 forms a contact with Tyr100 and Pro12 forms contacts with Arg97, His96, and Try100 of Mdm2 (Supplementary Table 1).

To examine the importance of different PMI side chains for the observed binding modes, we computed the change in binding free energy upon mutation $\Delta\Delta G$ but with the free energy of the associated state replaced by the free energy of macro-state $S_i$ (see Supplementary Note 3.2 and Supplementary Table 2). We observe that Phe3 and Trp7 are most important for stable binding. The role of the other side chains depends on the binding mode. For example, Thr1, Tyr6, Leu9, and Pro12 stabilize state 12 but not state 13. Alanine scanning experiments (Fig. 2e) have revealed that the Tyr6Ala mutant shows a similar $\Delta\Delta G$ to that of the Leu10Ala mutant even though the crystal structure shows no binding of Tyr6 to the inside of the hydrophobic cleft of Mdm2[59]. Our results thus suggest that the higher $K_d$ of the Tyr6Ala PMI mutant is not due to a destabilization of the crystal-like state, but may rather be explained by the destabilization of alternative bound states.

Other binding modes that involve more flexible PMI configurations do not strongly contribute to the binding affinity, but are relevant for the association process by "catching" PMI and funneling it into state 13. In the non-native states, PMI binds in different locations (3), in different orientations (5, 10, 11), or in unfolded conformations that dissociate relatively easily, but otherwise fold during the binding transition (4, 6, 8). The slowest transitions occur between states 12 and 11 and between states 13 and 7 that happen on milliseconds to hundreds of milliseconds. Non-native states that do not significantly contribute to binding pathways are briefly denoted as "trap". Trap states 5, 7, and 9 are predominantly reachable from state 13. Additional traps with lifetimes larger than 10 μs but not significant population were found, in which PMI binds far away from the binding site (structures not shown).

To resolve the dynamics inside the main binding pocket in greater detail, we split state 13 into the sub-states A–H with kinetics on time scales of a single microsecond or slower (Fig. 1 lower half and Supplementary Note 3.5). Sub-state A is structurally well-defined and contains the crystal structure (pdb code 3eqs), the crystallographically unresolved Pro12 forms contacts with Mdm2 Tyr100. Many of the sub-states (B, C, E, I)

are intermediates in the binding process (Fig. 3). In the crystal-like state (A) the Tyr6 side chain of PMI is not buried in the binding cleft. However in many non-native states, Tyr6 can either bind to the inner cleft together with Trp7 and Phe3 (D, B, H) or take the role of Trp7/Phe3 by anchoring PMI to the cleft (C, G, F and 5). Tyr6 can even take the place of Trp7 in a helically bound conformation that is similar to the crystallographic mode (E).

With the simulations conducted here, we find that state A has a stationary probability of 72%. Together states A and 12 have a joint stationary probability of 86%. Thus a large fraction of the strong affinity between PMI and Mdm2 is due to the two distinct but individually well-defined conformations 12 and A that interconvert directly on the timescale of 10 µs.

It is possible that the number of discovered non-natively bound structures, and their combined equilibrium probability, would continue to grow if the simulations would be extended. However, almost all metastable states found here are already visited in the first 60% of our simulation data (Supplementary Fig. 7) and the estimate for the binding free energy is converged (Fig. 2a). These indicators suggest that the non-natively bound structures with significant probabilities have been found.

**Binding mechanism**. To investigate the binding mechanism, we computed the reactive flux using transition path theory[36, 70] from the dissociated state to the crystal-like bound state (Supplementary Note 3.5). There are multiple parallel pathways and the metastable states can be grouped into on-pathway intermediates and off-pathway trap states—see Fig. 3 for an illustration of the major 60% of binding pathways. The most populous pathway (29%) goes through a partially folded state (C) that is anchored by Leu10 and Tyr6 to the binding cleft, while Phe3 and Trp7 form contacts with the outer surface of helix 2 of Mdm2. 15% of the reactive binding flux goes through states where PMI binds to the terminal region of the Mdm2 fragment that is located at the end of the binding cleft. A similar pathway was found for the p53-peptide in Ref. [71]. The terminally bound states form a conformational ensemble with various unfolded (not shown) and folded (3, I) PMI conformations. Among the terminally bound states, the macro-states that carry most reactive flux exhibit folded PMI. The folded conformations differ in the (hydrophobic) interface that they form with Mdm2 (3, I). Nine percent of the flux go through states 12 and E where PMI is almost in the crystal-like fold but the binding pattern is non-native. Inspection of the MD trajectories shows that during the fast transition from state E to the crystal-like state, the Tyr6 side chain leaves the binding cleft first and is then replaced by the Trp7 side chain all while Phe3 remains anchored to the cleft. In the transition between state 12 and the crystal-like state, the flexible C-terminus of PMI is rearranged such that Leu10 takes the place of Leu9 at the binding interface.

## Discussion

Multi-ensemble Markov models can be used to probe full multi-state kinetics of strong binders by combining conventional MD simulations of the binding process with biased MD simulations that spontaneously sample bound and unbound states. While standard analyses of enhanced sampling simulations do not readily provide kinetic information, MEMM estimators provide direct estimates of the kinetics without invoking macroscopic rate models. Using the nano-molar complex PMI–$^{25–109}$Mdm2 as an example, we obtained robust estimates of residence times that exceed the total amount of simulation data by three to four orders of magnitude and the individual simulation lengths by six to seven orders of magnitude.

Importantly, the inclusion of relatively little biased data enables us to sample rare events such as the protein-inhibitor dissociation steps, and drastically reduces the statistical error of rates and binding free energies compared to a MSM of purely unbiased MD data. In particular, we have demonstrated that MEMMs can effectively mitigate the problem of trajectories getting trapped in long-lived states. While direct estimation of MSMs requires that the visited states are reversibly connected—a condition that is difficult to test in high-dimensional systems—MEMMs only require irreversible visits to metastable states if those states were sampled reversibly in a biased simulation. On the other hand, in contrast to standard analysis methods such as WHAM or MBAR, MEMM estimators such as TRAM or TRAMMBAR do not require the full simulation data to be sampled from global equilibrium, thus greatly alleviating the sampling problem.

The binding/unbinding mechanism of PMI and Mdm2 was elucidated in full atomistic detail. While the binding is overall funnel-like, the detailed kinetics are quite complex. Rebinding can occur via multiple non-native intermediates on multi-milliseconds timescales. Another slow process is the inter-conversion of the crystallographic PMI–Mdm2 state with a newly identified state in which the C-terminus of PMI is unraveled and forms a new interaction pattern with Mdm2. Both states contribute significantly to the PMI–Mdm2 binding affinity and will inhibit binding to p53. The identification of such conformations gives us additional flexibility in optimizing the inhibitor.

Some minor trap states were found that do not significantly contribute to the binding affinity, but have lifetimes on the order of microseconds. Although such states may be overrepresented by current atomistic force fields[63], their existence implies that even for fast binders, around 100 µs of unbiased MD simulation are needed in order to characterize the association kinetics with statistical confidence.

The current study is a proof of principle—making optimal choice of starting structures and amount of data in unbiased vs. biased simulations depends on the molecular system, and a logical next step would be to make these choices iteratively within an adaptive sampling framework[28, 37, 72]. The present simulation approach makes progress towards the routine computation of residence times, and the identification of non-native or allosteric binding sites for protein-inhibitor systems. Because the approach does not require a priori knowledge of order parameters and structures, it can potentially be fully automated. With ever increasing computing power, this approach may become part of a high-throughput framework to compute protein–drug kinetics that may serve both pharmacological applications and the improvement of force fields towards the more accurate prediction of kinetics[73, 74].

MEMMs combine methods of free energy calculation and MSM estimation. Therefore any progress made in the development of protocols for free energy calculation might directly translate into a corresponding progress in the estimation of kinetics. We are confident that the seconds timescale is not the limit and that timescales comparable to the biological half-life of drugs (hours)[2], or the excessively long lifetimes of multivalent binders[17, 18] are, in principle, accessible. Equilibrium kinetic models of protein binding kinetics harvested with MEMMs can be embedded into particle-based reaction-diffusion simulations in order to probe the kinetics emerging from non-equilibrium conditions and the behavior of entire cellular signaling pathways[75].

## Methods

**MDM2-PMI simulation setup**. MD simulations were conducted with the Amber99SB-ILDN force field[76] and TIP3P water model[77] in the canonical

ensemble at temperature $T = 300$ K. To generate a starting structure, we used the heavy atom positions from the protein data bank (PDB) file 3eqs[57] and moved the peptide out of the binding pocket. Missing residues of the PDB structure (PMI Pro12 and Mdm2 Glu25) were modeled in standard conformations. Hydrogen atoms were added with AmberTools[78], the complex was solvated in a cubic box of edge length 7.62 nm with 13,698 water molecules, and five $Cl^-$ and one $Na^+$ counter ions were added. The two histidine residues of Mdm2 were protonated at the $\epsilon 2$ site. Simulations were performed with the ACEMD computer code[79] using the Langevin integrator using a damping constant $\gamma = 0.1$ $ps^{-1}$, constraints on the bonds that involve hydrogen atoms, and with heavy hydrogen atoms (four times the natural mass) to allow for an integration time step of 4 fs. Electrostatics were computed using Particle Mesh Ewald using a real-space cutoff of 0.9 nm.

**Hamiltonian replica-exchange simulations.** Since the relevant conformational states were a priori unknown, we avoided choosing structure-based collective variables but instead employed a so-called boost potential that was developed in the context of accelerated MD[80] and works by reducing the depth of the minima in the potential energy landscape. As the interaction of Mdm2 with PMI and other peptides is mostly hydrophobic[57, 81], the boost potential was applied to the Lennard-Jones interactions between the two chains and not the electrostatic interactions (see Supplementary Note 3.1 for simulation details). Six independent simulations starting from the crystallographic pose of about 1 µs length each were carried out with replica exchange[82] between 14 ensembles that interpolate between unbiased and strongly boosted potentials (Supplementary Note 3.1). The simulation took approximately $42 \times 10^3$ GPU hours.

**Unbiased MD simulations.** Short MD simulations of a total of 20 µs were used to explore the conformational space. From these simulations and all replicas of the replica-exchange simulations, starting conformations were uniformly sampled, generating various bound and unbound structures. In total, 502.597 µs of unbiased MD simulations were run. The initial structures were resolvated, energy minimized with 100 steps of conjugate-gradient descent, temperature equilibrated for 100 ps with harmonic constraints on protein and peptide atoms, followed by a 1 ns pressure equilibration with the Berendsen barostat. Finally the box size was set to the fixed cube with 7.62 nm edge length and an additional equilibration run of 1 ns was performed with active harmonic constraints. The production run generated 481 trajectories with varying lengths (between 945 and 1211 ns per trajectory). The simulation took approximately $115 \times 10^3$ GPU hours.

**TRAMMBAR is a new estimator for MEMMs.** Replica-exchange MD between different bias potentials can be extremely effective in exploring complex molecular state spaces[82]. Here we develop an extension of the bin-less TRAM method[49] to compute MEMMs in order to facilitate the integration of replica-exchange MD with unbiased MD. TRAM's ability to estimate unbiased kinetics relies on counting transitions between states within each simulation ensemble, but the contiguous simulation times between ensemble changes in a replica-exchange scheme are usually too short for that. We address the problem by splitting the data into two sets: (a) data from replica-exchange simulations for which we assume that it samples the equilibrium distributions of the respective ensembles and is thus analyzed with the MBAR framework[61, 62]; (b) data from unbiased MD simulations that are not long enough to sample the equilibrium distribution of the respective ensemble and are analyzed with bin-less TRAM. These two parts need to be coupled, and we call the resulting hybrid analysis method TRAMMBAR. Following Ref. [49], we denote the set of equilibrium samples (a) from ensemble $k$ by $X^k_{MBAR}$ and the set of time-correlated samples (b) from ensemble $k$ by $X^k_{TRAM}$. We approximate the reference equilibrium distribution as a point-wise distribution on all data by maximizing the likelihood

$$L_{TRAMMBAR} = L_{TRAM} \cdot L_{MBAR} \tag{1}$$

where $L_{TRAM}$ is defined as in Ref. [49]

$$L_{TRAM} = \prod_{k,i,j} \left( p^k_{ij} \right)^{c^k_{ij}} \prod_{\mathbf{x} \in X^k_{TRAM} \cap S_i} \mu(\mathbf{x}) e^{f^k_i - b^k(\mathbf{x})} \tag{2}$$

and $L_{MBAR}$ is the standard MBAR likelihood[61]

$$L_{MBAR} = \prod_k \prod_{\mathbf{x} \in X^k_{MBAR}} \mu(\mathbf{x}) e^{f^k - b^k(\mathbf{x})} \tag{3}$$

Here, $b^k(\mathbf{x})$ denotes the known unit-less bias energy of configuration $\mathbf{x}$ evaluated in the $k$th ensemble that can be obtained from the MD software. $c^k_{ij}$ are the observed transition counts from the time-correlated data $X^k_{TRAM}$, and $e^{-f^k} := \sum_i e^{-f^k_i}$ are the ensemble free energies. The likelihood is optimized by varying: the unbiased configuration weights $\mu(\mathbf{x})$, the joint equilibrium probabilities $e^{-f^k_i}$ to be in Markov state $S_i$ and ensemble $k$, and the transition probabilities $p^k_{ij}$, from which the kinetics at every ensemble can be computed. The TRAMMBAR algorithm is equivalent to the MBAR algorithm if $X_{TRAM}$ is empty, and equivalent to the TRAM algorithm

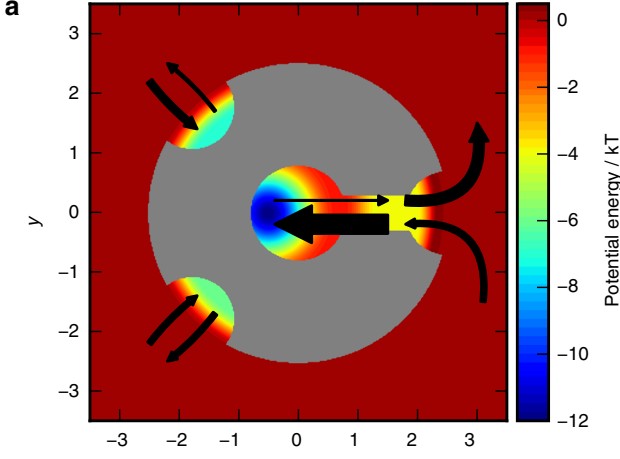

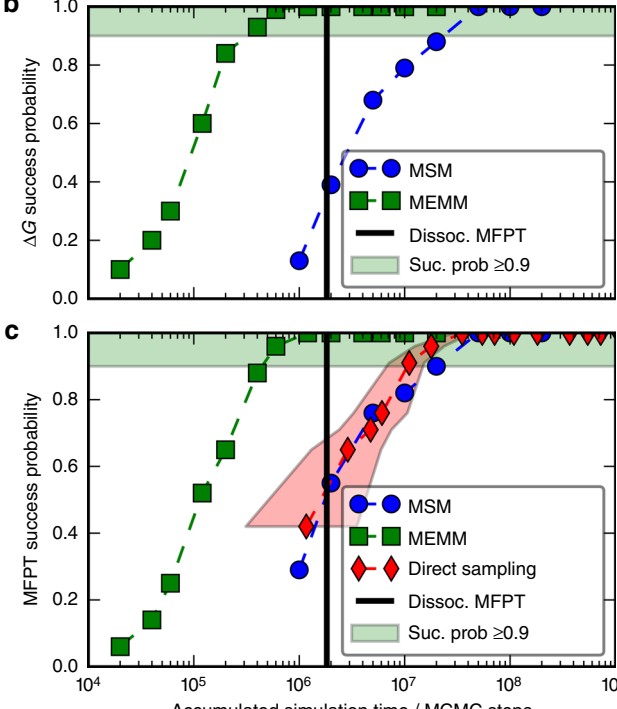

**Fig. 4** Illustration of computing rare-event kinetics with TRAMMBAR using a model for protein ligand binding. **a** Potential energy surface and transition rates between five states (bound, pre-bound, two mis-bound states, dissociated). Arrow thickness is proportional to rate. **b** Probability of computing the binding free energy $\Delta G$ within $1k_B T$ accuracy of the exact value for a given amount of simulation data using MEMMs (TRAMMBAR estimator) or MSMs. The vertical bar indicates the mean-first-passage time (MFPT) for dissociation. **c** Probability of computing the dissociation rate within factor $\frac{1}{2}$ to 2 accuracy of the exact value

with empty $X_{MBAR}$. The algorithm for maximizing the above likelihood is described in Supplementary Notes 1.1 and 1.2.

In order to illustrate our approach for computing rare-event kinetics for strong binders, consider the two-dimensional potential energy landscape in Fig. 4a. The gray shape represents a protein to which a small molecule ligand can bind. The protein has two shallow minima representing non-native binding sites on the surface, and a deep energy minimum representing an internal binding pocket at the end of a channel.

The mean dissociation time for this system is about $1.8 \times 10^6$ Monte Carlo steps (vertical bar in Fig. 4b, c). To approximate this time with direct simulation, multiple trajectories with lengths of at least $10^7$ steps need to be launched from the bound state (Supplementary Fig. 2). Using MSMs, still a total of $10^7$ steps of simulation time in shorter trajectories is needed to obtain accurate estimates of

both thermodynamics and kinetics (Fig. 4b, c). Using a MEMM with the TRAMMBAR estimator, we can get accurate estimates for both the binding free energy and the dissociation time with a total of only $5 \times 10^5$ steps that include short unbiased binding simulations and biased simulations on a flattened potential (Fig. 4b, c, Supplementary Note 2 and Supplementary Figs. 1 and 3). In contrast to most other enhanced sampling methods, a MEMM allows the computation of unbiased kinetics despite the fact that biases are used in the simulation. Moreover, MEMMs provide not only selected macroscopic rates, but full kinetics such as the whole set of transition rates shown in Fig. 4a.

**Multi-ensemble Markov model for Mdm2–PMI**. A MEMM was build from the MD and replica-exchange trajectories. To define MEMM states, we first chose the following set of features: all 1086 nearest-neighbor heavy atom distances between PMI residues and PMI residues (a) or Mdm2 residues (b) and the sine and cosine of the $\chi_1$ dihedral angle of Mdm2 Tyr100 (c), which is a known "gate-keeper" residue for ligand association[83]. The time-lagged independent component analysis (TICA) algorithm[84] with a lag time of 10 ns was used to obtain 20 independent components containing the slow kinetics. To these, trajectories of the minimal distance between PMI and Mdm2 were added to facilitate a clear definition of the fully dissociated state. The resulting feature trajectories were clustered with $k$-means ($k = 1000$). In total, 56 microstates discretizing the dissociated state were defined based on the minimal heavy atom distance between PMI and Mdm2 and added to the set of the 1000 $k$-means clusters. The dissociated states had to be defined explicitly because of the low metastability of the dissociated state in the simulation box which prevents that the TICA algorithm finds a dimension that describes the full association/dissociation process of the binding partners (see Supplementary Fig. 10 for the influence of the definition of the dissociated states on the estimates of the binding free energy and of the residence time). Transition counts were computed for TRAMMBAR and for the MSM. For TRAMMBAR the initial 50 ns of the replica-exchange trajectories were discarded and the rest was subsampled, taking only one frame every 0.1 ns. We picked a lag time of 150 ns for TRAMMBAR based on the convergence of the implied time scales and mean-first-passage-times (Supplementary Notes 3.4 and 3.7, Supplementary Figs. 5 and 8). All analyses were done using PyEMMA[60] and MDTraj[85].

**Experimental binding kinetics**. The association and dissociation rate measurements were performed in stopped-flow and competition fluorescence anisotropy experiments. For the association measurements, FITC-PMI and Mdm2 were rapidly mixed using an SX20 stopped-flow spectrometer (Applied Photophysics). The temperature was maintained at 25 °C, and an excitation wavelength of 493 nm, in conjunction with a 515 nm long-pass filter was utilized. For the dissociation measurements, 10 nM FITC-PMI peptide was incubated with Mdm2, then excess of the unlabeled PMI (10 µM) was added and the dissociation was followed with a Multilabel 384-well plate reader (Tecan, Infinite M1000 PRO) with excitation at 494 nm and emission at 517 nm (see Supplementary Notes 4.1–4.3 and Supplementary Figs. 11 and 12 for details).

**Code availability**. TRAMMBAR has been implemented in the PyEMMA software. PyEMMA is available free of charge at http://pyemma.org.

**Data availability**. The molecular dynamics data that support the findings of this study are available in the Edmond Open Access Data Repository with the identifier doi:10.17617/3.x.[86] All relevant data is available from the authors upon request.

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

## Acknowledgements

We thank Matt Harvey (Acellera Ltd.) and Markus Rampp and Ingeborg Weidl (Max Planck Computing and Data Facility) for technical support and the Max Planck Society for usage of the Hydra supercomputer. We thank the group of Prof. Tad Holak in the Jagiellonian University, Poland for assistance with the Mdm2 purification and refolding protocols. We thank Prof. Sebastian Springer in Jacobs University Bremen for the usage of plate reader (Tecan, Infinite M1000 PRO). We thank Dr. S.L. Shammas, who kindly provided the script used to analyze the association kinetics data. We are grateful to Kresten Lindorff-Larsen, Vincent A. Voelz, John D. Chodera as well as all members of the Computational Molecular Biology group for insightful discussions. Funding is acknowledged by European Commission (ERC StG "pcCells" to F.N.), Deutsche Forschungsgemeinschaft (SFB 1114/C3, SFB 958/A4, SFB 740/D7, and TRR 186/A12 to F.N. and SFB 1114/A4 to F.N. and T.R.W.). J.C. is a Wellcome Trust Senior Research Fellow (WT 095195MA). J.S. is a Marie Skłodowska-Curie Internationally outgoing fellow. M.D.C. is supported by a Biotechnology and Biological Sciences Research Council (BBSRC) studentship.

## Author contributions

F.P., C.W., E.T.A., T.R.W. and F.N. have designed research. F.P. and H.W. have developed methods. F.P. and C.W. have developed/implemented software. F.P. and C.W. have conducted simulations. F.P. and C.W. have analyzed simulations. E.T.A., C.F. and J.C.

have designed experiments, E.T.A., M.D.C. and J.S. have conducted experiments. F.P., C.W., E.T.A., and F.N. have written the paper.

## Additional information

**Competing interests:** The authors declare no competing financial interests.

