## [Peer Review File · Nature Communications]

Reviewers' comments:

Reviewer #1 (Remarks to the Author):

The authors present a model of the binding/unbinding of a protein fragment with a peptide inhibitor. Impressively, their model captures seconds timescales by exploiting a new multi-ensemble MSM method recently introduced by the authors. The model provides interesting insights, such as the existence of multiple bound poses. I expect this paper to have significant impact since it is the first demonstration of the power of this new method and provides interesting biophysical insights. It would benefit from clarifying a number of points, raised below.

- The authors claim that the pharmaceutical industry has shifted its focus to optimizing drug-target kinetics. I think it would be more accurate to say that there is growing interest in the potential relevance of this kinetics and methods like those the authors present could be useful for achieving this end. There still seems to be significant controversy over the importance of kinetics.
- I expected a larger experimental component to the paper based on the abstract, but really there's only one new parameter measured. I recommend saying something like "We tested our model against existing mutagenesis data and our own experimental measurements of the unbinding rate."
- Presumably there's a chance the model wouldn't work, so it would be more accurate to talk about "testing" the model instead of "validating" it. The model is only validated if the test is successful.
- It would be more accurate to cite Voelz et al. JACS 2010 for being able to reach the millisecond timescale with distributed computing than the Shirts et al. paper.
- Its worth mentioning the simulation method and total aggregate simulation timescale for the Zwier et al. work as a reference point for judging how much of an improvement the authors obtained with their own approach.
- Please run spell-check, there are many typos, such as "using using the MBAR estimator" (duplicates "using").
- "Hamilton" replica exchange should be "Hamiltonian" replica exchange.
- Why do you only observe 11 binding events (5 from unbiased simulations and 6 from the biased ones) but almost 60 unbinding events if binding is so much faster? How is the MSM arriving at such large binding rates?
- Its not clear when you're using your MSM and when you're using more ad hoc analyses of the

raw simulation data. For example, I assumed you got the binding free energy from your MSM, but then it sounds like you just used a distance cutoff between the two peptides to define bound and unbound states. It also sounds like you called things bound if any pair of heavy atoms from the two peptides is within 1 nM. That sounds like an unrealistically generous definition of what it means to be bound. Please clarify whether you used an MSM or something else for each of the main results in the paper. If you're using a distance cutoff, it would also help to give some analysis of how much the results change as you vary the cutoff. If you're using a distance cutoff instead of the MSM, I'd also like to know why?

- Why are you satisfied with your dissociation constant being off by a factor of ten?

- I'm a bit alarmed that you switch force fields and water models for your perturbation analysis. It also sounds like you used a distance cutoff to define the bound state. Why not use the definition of states from your MSM?

- In the caption of Fig. 2, please explain what the axes of panels a-d are. In panel a, does 50% mean half of the unbiased data and half of the biased data? In panel c, are you using all of the unbiased data and varying how much of the biased data you use? Or are you using a fixed amount of simulation but varying how much of that simulation data comes from biased/unbiased simulations?

- You claim that your association rate is similar to those for other peptides but don't give enough information for the reader to evaluate that claim. Please discuss how those peptides differ from the one you simulated and what their association kinetics are.

- How was the MEMM coarse-grained? I thought you used PCCA, but then it sounded like you defined the unbound state based on a cutoff. How did you test the coarse-grained model was reasonable (I'm not sure which model Fig. S7 is for)? Using 14 states isn't an obvious choice from the implied timescales and its not clear how you chose 83 substates for state 13.

- The conformation you show for state 12 doesn't look folded, assuming state A is representative of the folded state.

- Why do you make a point of saying PMI mostly has an RMSD less than 1 nM? That's a huge RMSD, so I don't know what you're trying to convey.

- What does "resolving the stability changes of alanine mutations by state" mean?

- In the paragraph on the substates of state 13, it would be good to tell the reader why you discuss these specific residues (Tyr6, Trp7, Phe3).

- Are the state labels in Fig. 3 the same as Fig. 1? It kind of sounds like you used a distance cutoff to define the crystal-like state again, which lead to some confusion over what your state definitions are.

- For adaptive sampling, it would also make sense to cite the related method from Zimmerman and Bowman. JCTC 2015.

Reviewer #2 (Remarks to the Author):

The manuscript by Paul, Wehmeyer et al is a thorough application of new ideas from markov state model (MSM) and related approaches applied to a complex problem. Apart from the obvious biological importance, another feature which in my view make this paper worthy of publication in Nature Communications, is that it demonstrates cleanly how to mix two popular classes of methods, enhanced sampling and MSM.

However before I recommend publication I have some concerns which I would like to see addressed.

(1) Fig. 2 is of profound importance to the simulation community. On one hand it gives immense credibility to this work (to a partial extent even addressing my concern #2). On the other hand it makes me very worried about the innumerable papers published over the years using straightforward MSM. From what the authors show here, without mixing information from enhanced sampling, MSM can be completely misleading for systems with rare events, with error bars spread across several orders of magnitudes. I find this disturbing for the many vanilla-MSM papers out there estimating kinetics for rare event systems, including drug binding. I would request the authors to critically address this issue.

(2) The protocol here, of doing replica exchange with protein-ligand interactions tuned will give accurate rate matrices and rates *if and only if* the replica exchange scheme has actually sampled states per the equilibrium distribution. This in turn will be true if ligand-protein interactions are the only slow degree of freedom, or if all other slow degrees of freedom are sufficiently coupled to this particular degree of freedom. But that might not be true, and thus contrary to what authors claim in the introduction and in the conclusion, they still have a collective variable problem. If the drug binding is actually limited by some other degree of freedom - say protein conformational transitions or solvation of binding pocket/some other residues, I fail to see why the MSM constructed on the basis of the simple replica exchange scheme these authors do, would be accurate. Fig 2 might actually be indirect evidence, necessary but still not sufficient, that for this system that the authors got lucky with just using protein-ligand interactions. In summary, I think the authors should re-examine their claim of not needing knowledge of collective variables. I don't think its true. Replica exchange if not done carefully suffers from poor sampling issues, and this is the reason for existence of methods like OSRW by Wei Yang and co-workers, and REST by Berne and co-workers.

(3) In the abstract and elsewhere the authors mention that the binding comes from interchange between different conformations with different hydrophobic surfaces. Were all these conformations visited in the original 300 or 500 microsecond MD? Would there have been more

conformations if the original MD was run 5000 microseconds, or 50,000 microseconds? How can the authors be sure about the sensitivity to this parameter? realize the authors compare the statistics from using 300 and 500 microsec runs, but if there are unvisited conformations even in 500 micro sec, this might not be sufficient. It would be nice to see a brief discussion on this point.

(4) What can the authors say about the unbinding mechanism instead of the binding mechanism, which (the former) is often of great pharmacological relevance? With how much confidence can they say this?

(5) How much wall clock time did the original 500 microsecond MD take? Please mention this in the main text as I think this is an important aspect for practical applications.

In summary, I am happy with the paper but would like to see these few concerns alleviated before I can recommend publication.

Reviewer #1

Comments: The authors present a model of the binding/unbinding of a protein fragment with a peptide inhibitor. Impressively, their model captures seconds timescales by exploiting a new multi-ensemble MSM method recently introduced by the authors. The model provides interesting insights, such as the existence of multiple bound poses. I expect this paper to have significant impact since it is the first demonstration of the power of this new method and provides interesting biophysical insights. It would benefit from clarifying a number of points, raised below.

We thank the referee for the supporting assessment.

- *The authors claim that the pharmaceutical industry has shifted its focus to optimizing drug-target kinetics. I think it would be more accurate to say that there is growing interest in the potential relevance of this kinetics and methods like those the authors present could be useful for achieving this end. There still seems to be significant controversy over the importance of kinetics.*

We agree and have adapted the abstract and introduction in the revised manuscript accordingly.

- *I expected a larger experimental component to the paper based on the abstract, but really there's only one new parameter measured. I recommend saying something like "We tested our model against existing mutagenesis data and our own experimental measurements of the unbinding rate."*

Thank you for this suggestion. We have adapted the abstract to make this more explicit. Moreover we now have extended the experimental component of the paper to also include measurements of the association rate constant.

- *Presumably there's a chance the model wouldn't work, so it would be more accurate to talk about "testing" the model instead of "validating" it. The model is only validated if the test is successful.*

We have replaced "validate" by "test" where appropriate in the revised manuscript.

- *It would be more accurate to cite Voelz et al. JACS 2010 for being able to reach the millisecond timescale with distributed computing than the Shirts et al. paper.*

Thank you for pointing out this inaccuracy. We have clarified that this paragraph refers to all-atom (explicit solvent) simulations and we have updated the references to cited papers that report notable applications of millisecond-simulations with explicit solvent. Please let us know if we have missed something that was published earlier.

- *Its worth mentioning the simulation method and total aggregate simulation timescale for the Zwier et al. work as a reference point for judging how much of an improvement the authors obtained with their own approach.*

We have added a corresponding sentence to the manuscript (120 μ s of implicit solvent MD).

- *Please run spell-check, there are many typos, such as "using using the MBAR estimator" (duplicates "using").*

Thank you! This has been corrected in the revised manuscript.

- *"Hamilton" replica exchange should be "Hamiltonian" replica exchange.*

This has been corrected in the revised manuscript.

- *Why do you only observe 11 binding events (5 from unbiased simulations and 6 from the biased ones) but almost 60 unbinding events if binding is so much faster? How is the MSM arriving at such large binding rates?*

The large number of dissociation events can be explained by the fact that we started all replica-exchange simulations in the crystallographic pose (this is now mentioned in the revised main text). If the complex had dissociated in all ensembles and all repeats, this would have resulted in at least $N_{\text{repeats}} \cdot N_{\text{ensembles}} = 6 \cdot 14 = 84$ unbinding events. Unbinding is only a frequent event in the biased simulations, so there is no contradiction with a large binding rate in the unbiased ensemble (whose kinetics are described by the MSM).

However, the number 58 might be misleading, since we exclude the first 50 ns of the biased simulations from the subsequent analysis. The rest of biased data ($t \geq 50$ ns) still contains 26 full unbinding events and 6 full binding events (from completely dissociated to the crystallographically bound pose). The revised text now reports these numbers.

- *Its not clear when you're using your MSM and when you're using more ad hoc analyses of the raw simulation data. For example, I assumed you got the binding free energy from your MSM, but then it sounds like you just used a distance cutoff between the two peptides to define bound and unbound states...*

In this manuscript, there is only one instance of ad hoc-analysis and it is reported in the sentence: "Inspection of the MD trajectories shows that during the fast transition from state E to the crystal-like state, the Tyr6 side-chain leaves the binding cleft first and is then replaced by the Trp7 side chain all while Phe3 remains anchored to the cleft." All other analyses are based either on the MSM that was estimated from all simulation data or on the biased simulation data alone (in which case the analysis was done with MBAR).

We have revised the manuscript to clarify when raw data analyses and MSM analyses were used, and to clarify the definition of the states (see methods section "Multi-ensemble Markov model for Mdm2-PMI" and supplementary section "Coarse-graining and TPT analysis"). The dissociated state is consistently defined in the MSM and in all other analysis as the set of conformations with a minimal heavy-atom distance larger than 1 nm between PMI and Mdm2.

- *... It also sounds like you called things bound if any pair of heavy atoms from the two peptides is within 1 nm. That sounds like an unrealistically generous definition of what it means to be bound. Please clarify whether you used an MSM or something else for each of the main results in the paper...*

We agree that a 1 nm cutoff is too generous to define the bound state from a chemical standpoint. In the original manuscript, we had used the term "bound" to include all states that are not dissociated (including non-natively associated states), as our fluorescence experiments are not able to distinguish between different associated substates. To avoid confusion we are now calling these states "associated" in the revised manuscript. This state is consistently defined as the set of conformations with a minimal heavy-atom distance less of equal than 1 nm between PMI and Mdm2 for all analyses (MSM or raw data). Otherwise we speak of natively or crystal-like bound and non-native bound structures, where appropriate.

- *... If you're using a distance cutoff, it would also help to give some analysis of how much the results change as you vary the cutoff...*

To address this request, we now test the influence of the distance cutoff d that defines the dissociated state upon our estimates of the free energy of binding and for the residence time – see Supplementary Figure 10. We find that the results depend on the choice of the cutoff but all variation is within the range of the statistical errors of the numbers that are reported in the main text.

- *... If you're using a distance cutoff instead of the MSM, I'd also like to know why?*

Note that the dissociated state is still treated as an MSM state, so it is not used *instead* of an MSM. The reason to define the dissociated state explicitly instead of just relying on the results of data clustering is that the dissociated state in the given simulation box has a very low metastability.

Thus, within the simulated system, the binding process is not a slow process. This prevents the TICA method from finding independent components (i. e. order parameters) that describe the full association/dissociation process of the binding partners, and a blind clustering on that space would not result in a cleanly separated dissociated state. In order to resolve the binding/unbinding process in the MEMM, we thus chose this cutoff as a help to the clustering method. A similar procedure was applied in an earlier work [Plattner and Noé, *Nat. Commun.* **6**, 7653 (2015)].

- *Why are you satisfied with your dissociation constant being off by a factor of ten?*

When comparing two sets of probabilities or rates, an order-of-magnitude agreement corresponds to about 2.3 kT or 1.4 kcal/mol accuracy in energies, and is what can be expected *on average* when comparing current force-fields with each other and force-fields with experiments. Similar limitations to accuracy can sometimes also be found between different experimental setups. Although better agreement can occur coincidentally, there is evidence for the order-of-magnitude agreement hypothesis:

Works by Best *et al.* [*J. Chem. Theory Comput.* **10**, 5113 (2014)] indicate that current force fields overestimate the binding between proteins. In [Petrov, Zagrovic, *Plos Comput. Biol.* **10**, e1003638 (2014)] this effect is also seen for the amberSB99-ILDN force field that we use in our work. Best *et al.* report a K_d value for Villin HP36 that is at least 10 times underestimated, so a factor of 10 might actually be what to expect. However this is further complicated by the fact that current force fields might not correctly represent the unbound structure of PMI [Rauscher *et al.*, *J. Chem. Theory Comput.* **11**, 5513 (2015)] which could lead to an erroneous entropic contribution to the free energy of binding.

A comparable work about protein-peptide binding might be [Do *et al.*, *J. Chem. Theory Comput.* **12**, 395 (2016)], a 6 μ s metadynamics study of protein-IDP (intrinsically disordered protein) binding. The authors find deviations of 2.6 to 3.8 kcal/mol between experiment and simulation which however might also be due to lack of sampling, which would correspond to a factor of 75 to 560 in probabilities. Also in [Yuwen *et al.*, *Biochemistry*, **53**, 6473 (2014)], the authors find a hundredfold deviation of K_d between experiment and simulation.

We have added references to current works on force field validation to the discussion section in the manuscript.

- *I'm a bit alarmed that you switch force fields and water models for your perturbation analysis. It also sounds like you used a distance cutoff to define the bound state. Why not use the definition of states from your MSM?*

We have to switch force field for technical reasons. The mutations to alanine are modeled by replacing the corresponding side-chain of the amino acid by a hydrogen in the MD trajectories. The removal of the side chain would leave an energetically unfavorable “hole” in explicit-solvent models. For this reason explicit-solvent “perturbations” would only be feasible if we reran simulations for the mutants, which is computationally unfeasible. In the implicit solvent model the void introduced by the removed side chain is modeled automatically as “water” (i. e. as a region of high dielectric constant), therefore a perturbation analysis makes sense when combining an ensemble of wild-type *solute* structures with an *implicit solvent* model.

The implicit solvent force field that we use has been shown to be accurate for structure prediction of folded proteins. A validation study by Zeller and Zacharias [*J. Phys. Chem. B* **118**, 7467 (2016)] shows that implicit solvent force fields can perform comparably to explicit solvent force fields in predicting binding free energies, however we only predict changes in binding free energy, which is a simpler task. We have added this explanation to the revised SI (section “Mutation model”).

We in fact use the same definition of the bound state (“associated state” in the current manuscript) in the MSM and in the perturbation analysis. The dissociated macro-state is consistently defined as all conformations with $d > 1.0$ nm in all analyses (See also SI section “Coarse-graining and TPT analysis”).

- *In the caption of Fig. 2, please explain what the axes of panels a-d are. In panel a, does 50% mean half of the unbiased data and half of the biased data? In panel c, are you using all of*

the unbiased data and varying how much of the biased data you use? Or are you using a fixed amount of simulation but varying how much of that simulation data comes from biased/unbiased simulations?

We have extended the caption of Fig. 2 to clarify these points. Indeed in panels a,b 50% means 50% of biased and 50% of unbiased data. In panels c,d we are varying the amount of biased data between 0% and 100% of all biased data. The amount of unbiased data is varied too, such that every estimate is computed from exactly 502 μs of data (biased + unbiased).

- *You claim that your association rate is similar to those for other peptides but don't give enough information for the reader to evaluate that claim. Please discuss how those peptides differ from the one you simulated and what their association kinetics are.*

For the revised version of the manuscript we have conducted stopped flow experiments to measure the association rate of PMI to the Mdm2 fragment that was used in the simulation. We find $k_{\text{on}}^{\text{exp}} = 5.27 \cdot 10^8 \text{ M}^{-1}\text{s}^{-1}$ that is within an order-of-magnitude agreement with the simulation result $k_{\text{on}}^{\text{sim}} = 3 \cdot 10^9 \text{ M}^{-1}\text{s}^{-1}$ (see main text of the manuscript for error estimates and discussion).

Our claim that the association rate is similar to those for other peptides is actually not justified by the literature reference that we gave in the unrevised version of the manuscript. We have corrected this error in the manuscript. The literature data for the experimentally measured association rates of other p53-peptides to Mdm2 in [Schon et al, *J. Mol. Biol.* **323**, 491 (2002)] refers to the whole N-terminal domain (amino acids 2 to 125) of Mdm2 while we only simulated amino acids 25 to 109 of Mdm2. The first 24 amino acids of Mdm2 form a lid that competes for the binding cleft [Showalter et al., *J. Am. Chem. Soc.*, **130**, 6472 (2008)]. As expected, the association rates reported by Schon et al. (on the order of $10^7 \text{ M}^{-1}\text{s}^{-1}$) are significantly lower than those to the $^{25-109}\text{Mdm2}$ fragment.

We have added a reference to the simulation study by Zwier et al. [*J. Phys. Chem. Lett.* **7**, 3440 (2016)]. They report a rate $k_{\text{on}} = 7 \cdot 10^7 \text{ M}^{-1}\text{s}^{-1}$ for association of the peptide $^{17-29}\text{p53}$ to $^{25-109}\text{Mdm2}$ which is roughly a factor ten slower than the association of PMI to $^{25-109}\text{Mdm2}$ ($k_{\text{on}}^{\text{exp}} = 5.27 \cdot 10^8 \text{ M}^{-1}\text{s}^{-1}$).

- *How was the MEMM coarse-grained? I thought you used PCCA, but then it sounded like you defined the unbound state based on a cutoff. How did you test the coarse-grained model was reasonable (I'm not sure which model Fig. S7 is for)? Using 14 states isn't an obvious choice from the implied timescales and its not clear how you chose 83 substates for state 13.*

Please let us clarify first that all quantities except for the transition rates in main text Fig. 1 are computed from the full MSM with 1056 microstates. This includes ΔG , k_{off} and the TPT fluxes in main text Fig. 3. The coarse-grained transition matrix is solely used for visualization of the transition network and is never used to extrapolate to long lag times ($\tau > 150 \text{ ns}$). The 1056-state MEMM was validated by the Chapman-Kolmogorov test shown in Suppl. Figure 6.

All macro-states except for the unbound macro-state were determined with the PCCA algorithm (see SI "Coarse-graining and TPT analysis"). The unbound macro-state (called "dissociated macro-state" in the revised manuscript) is defined as the set of all dissociated microstates (minimal PMI-Mdm2-distance larger than 1 nm). We have added an explanation of how and why the dissociated macro-state is defined to the SI (section "Coarse-graining and TPT analysis").

Since the coarse-grained MSM is never used to compute association or dissociation rates, the number of macro-states is arbitrary. Judging from the implied time scale estimates of the full MEMM, 13 is the number macro-states that interconvert on the 10 μs timescale (and slower) and 83 is the number of states that interconvert on the microsecond time scale (and slower).

- *The conformation you show for state 12 doesn't look folded, assuming state A is representative of the folded state.*

Indeed, state 12 is not folded but only shows one helical turn from Phe 3 to Tyr 6. We have corrected the corresponding parts of the manuscript.

- *Why do you make a point of saying PMI mostly has an RMSD less than 1 nM? That’s a huge RMSD, so I don’t know what you’re trying to convey.*

What we wanted to say is that the most probable macro-states are states 12 and A that are structurally well defined and contribute a large fraction to the binding affinity. We have revised the corresponding paragraph in the main text and do no longer refer to the RMSD. We have removed Suppl. Figure 5, since we do not need it to support our argument.

- *What does "resolving the stability changes of alanine mutations by state" mean?*

Apologies for the confusing statement. We have clarified this sentence and added a longer explanation to the SI. We compute the change in binding free energy upon mutation, $\Delta\Delta G$, but with the free energy of the bound state replaced by the free energy of PCCA state S_i . That is, we pretend that the only bound state of PMI and Mdm2 is the PCCA state S_i and ask how then the binding free energy would change if some PMI side chain was mutated to alanine. This is a way of measuring the effect of mutations on the stability of the PCCA states.

- *In the paragraph on the substates of state 13, it would be good to tell the reader why you discuss these specific residues (Tyr6, Trp7, Phe3).*

The amino acids Phe3, Tyr6, Trp7 and Leu10 are the most important residues important for PMI-Mdm2 binding. The role of Phe3 of Trp7 and Leu10 in binding is rather clear because they form the binding interface in the crystal structure (pdb ID 3eqs). Alanine scanning experiments by Li *et al.* [*J. Mol. Biol.* **398**, 200 (2010)] revealed that Tyr3Ala PMI mutant shows a similar $\Delta\Delta G$ to that of the Leu10Ala PMI mutant even though the crystal structure shows no binding of Tyr6 to the inside of the hydrophobic cleft of Mdm2. Our results indicate that the higher K_d of the Tyr6Ala PMI mutant could possibly be explained by the destabilization of alternative bound states (e. g. macro-state 12) that are distinct from the crystal-like state (A).

We have added this explanation to the main text of the manuscript.

- *Are the state labels in Fig. 3 the same as Fig. 1? It kind of sounds like you used a distance cutoff to define the crystal-like state again, which lead to some confusion over what your state definitions are.*

Yes, the labels are identical. We have changed the label “unbound” to “dissociated” and have changed “crystal-like” to “crystal-like (A)” in the revised manuscript to further clarify the identity of the states.

In the unrevised manuscript we had defined the target state for transition path theory (Fig. 3) to encompass all microstates that have a (mean) RMSD less than 0.3nm to the crystal structure. For simplicity, we have now replaced this definition by using state A, since it also contains the crystal-like conformations. This change has no relevant effect on the results: the qualitative picture of the binding mechanism stays the same. Only the relative magnitudes of the reactive fluxes change slightly. We have adapted Fig. 3 and its discussion in the revised manuscript accordingly.

- *For adaptive sampling, it would also make sense to cite the related method from Zimmerman and Bowman. JCTC 2015.*

Thank your for this suggestion, we have added the literature reference to the manuscript.

Figure 1: Fig. 3 from the manuscript, influence of the definition of the TPT end-state on the results. blue: original numbers from the unrevised manuscript, green: new TPT calculation where the previous low-RMSD end-state was replaced by A.

Reviewer #2:

Comments: The manuscript by Paul, Wehmeyer et al is a thorough application of new ideas from Markov state model (MSM) and related approaches applied to a complex problem. Apart from the obvious biological importance, another feature which in my view make this paper worthy of publication in Nature Communications, is that it demonstrates cleanly how to mix two popular classes of methods, enhanced sampling and MSM.

However before I recommend publication I have some concerns which I would like to see addressed.

1. *Fig. 2 is of profound importance to the simulation community. On one hand it gives immense credibility to this work (to a partial extent even addressing my concern #2). On the other hand it makes me very worried about the innumerable papers published over the years using straight-forward MSM. From what the authors show here, without mixing information from enhanced sampling, MSM can be completely misleading for systems with rare events, with error bars spread across several orders of magnitudes. I find this disturbing for the many vanilla-MSM papers out there estimating kinetics for rare event systems, including drug binding. I would request the authors to critically address this issue.*

We agree with the referee that MSMs *can* be completely misleading if not properly analyzed, as there are cases where the MSM appears “reversible connected”, but the underlying data has not truly interconverted between the metastable states. In addition, we think that the problem of incorrect data treatment is even more prevalent in vanilla enhanced sampling simulations that are often analyzed with methods that assume global equilibrium (e.g. WHAM or MBAR), which is even more difficult to achieve than reversible connectivity. While for both these problems, statistical indicators exist that should be employed in a thorough analysis, these sampling problems were exactly the motivation to develop methods such as MEMMs. To discuss this point, we have now added the following paragraph to the Conclusion section:

“In particular, we have demonstrated that MEMMs can effectively mitigate the problem of trajectories getting trapped in long-lived states. While direct estimation of MSMs requires that the visited states are reversibly connected – a condition that is difficult to test in high-dimensional systems –

MEMMs only require irreversible visits to metastable states if those states were sampled reversibly in a biased simulation. On the other hand, in contrast to standard analysis methods such as WHAM or MBAR, MEMM estimators such as TRAM or TRAMMBAR do not require the full simulation data to be sampled from global equilibrium, thus greatly alleviating the sampling problem.”

With MSMs, it is in general difficult to decide whether the trajectories that are used for estimating the MSM are reversibly connected, in other words whether there’s a path connecting all pairs of states. This question is actually undecidable unless a single ultra-long simulation trajectory was used, because separate trajectories will never visit the exact same points in configuration space and will thus always separate into disconnected sets if a fine enough discretization would be used. For example, if a peptide dissociates from the protein in one simulation and binds to it in another, this may indicate connectivity, but do they really bind in the same conformation? This is a question of resolution. There are at least two ways to deal with the MSM disconnectivity problem in practice: (i) one could use a variant of adaptive sampling to start from one state A and extend MD trajectories until the other state B is reached (see [Plattner *et al.*, *Nat. Chem.* DOI: [10.1038/nchem.2785](https://doi.org/10.1038/nchem.2785) (2017)]). Doing this both ways guarantees connectivity of A and B . (ii) We have advocated statistical tests such as the Chapman-Kolmogorov tests that can indicate problems with connectivity [e.g., Prinz *et al.*, *J. Chem. Phys.* **134**, 174105 (2011)]. Also checking the convergence of the stationary distribution as the lag time is increased is an useful approach to spot connectivity problems (see [Nüske *et al.*, *J. Chem. Phys.*, **146**, 094104 (2017)] and [Doerr and De Fabritiis, *J. Chem. Theory Comput.*, **10**, 2064 (2014)]).

We have added the following paragraph to the introduction section to discuss this point: “*While MSMs help with parallelizing this problem and rare events can be sampled, in particular when adaptive sampling strategies are combined with high-throughput simulation, the sampling of very rare events such as protein-inhibitor dissociation can still be very inefficient. In practice, this difficulty may result in not properly connected models and underestimated or imprecisely estimated residence times. While MSM analyses have the advantage of being able to detect these problems with carefully conducted Markovianity tests and by computing binding free energies as a function of the MSM lag time, the typical solution involves running more simulations, which is unpractical when computational resources are limited.*”

2. *The protocol here, of doing replica exchange with protein-ligand interactions tuned will give accurate rate matrices and rates *if and only if* the replica exchange scheme has actually sampled states per the equilibrium distribution. This in turn will be true if ligand-protein interactions are the only slow degree of freedom, or if all other slow degrees of freedom are sufficiently coupled to this particular degree of freedom. But that might not be true, and thus contrary to what authors claim in the introduction and in the conclusion, they still have a collective variable problem. If the drug binding is actually limited by some other degree of freedom - say protein conformational transitions or solvation of binding pocket/some other residues, I fail to see why the MSM constructed on the basis of the simple replica exchange scheme these authors do, would be accurate. Fig 2 might actually be indirect evidence, necessary but still not sufficient, that for this system that the authors got lucky with just using protein-ligand interactions. In summary, I think the authors should re-examine their claim of not needing knowledge of collective variables. I don't think its true. Replica exchange if not done carefully suffers from poor sampling issues, and this is the reason for existence of methods like OSRW by Wei Yang and co-workers, and REST by Berne and co-workers.*

We agree with you in the statement, that indeed we use a collective variable for the HREMD simulations. However we want to convey the fact that *in general* the TRAM/MEMM framework *does not* rely on good choices of reaction coordinates, because all variables can be treated in an MSM-way a posteriori (application of TICA, discretization, estimation of conditional jump probabilities). If the replica simulations had been conducted with less frequent exchanges, or even without exchanges, the data could even have been analyzed with the original TRAM procedure [Wu *et al.*, *Proc. Natl. Acad. Sci. USA* **113**, E3221 (2016)] which does not require that the biased simulations sample from the global equilibrium distribution. In this analysis framework, the choice of collective variables used in

the enhanced sampling only affects how efficient or helpful these are in terms of sampling the rare events, but they will not bias the results.

We have added the following, clarifying paragraph in the results section “Multi-ensemble Markov models reveal fast binding and slow dissociation kinetics”:

“While TRAM requires all simulations to be longer than its lag time (often on the order of tens to hundreds of nanoseconds), this is not the case for replica-exchange simulations with rapid exchanges. TRAMMBAR can employ such replica-exchange data, by assuming global equilibrium for that part of the simulation, which is justified when statistical tests indicate short correlation times. The present replica-exchange data has a correlation time of 40 ns, compared to simulation lengths of about 1 μ s (Supplementary Information).”

In order to give the referee some more insights why we believe that the current HREMD protocol does indeed sample from global equilibrium, please consider the following arguments:

- *The bias potential was chosen carefully.* We tested various ways of biasing the PMI-Mdm2 interaction including multiple ways of scaling the Lennard-Jones and the electrostatic interaction energy between the binding patterns. Scaling of some terms of the Hamiltonian is very similar to the REST1/2 methods. Despite long parameter optimization, we were not able to see reversible binding and unbinding in the biased replica trajectories (continuous in coordinates). Only changing the functional form of the bias from a scaled interaction potential to the boost potential resulted in reversible binding/unbinding.
- *We quantified the relaxation speed to equilibrium and sampling efficiency.* We computed the statistical inefficiency as defined in [Shirts, Chodera. *J. Chem. Phys.* **129**, 124105 (2008)] for the HREMD simulations. We find 39.8 ns (95% confidence interval [5.2 ns, 184.0 ns]) which is much smaller than 7.3 μ s, the total simulation length per replica. This number is related to the autocorrelation time of the system and gives the length of the time interval after which two samples can be considered as uncorrelated. We computed this number from the time series of the microstate indices (i. e. discrete trajectories), therefore it should be sensitive to all the slow processes that are resolved in the MSM.
- *Our HREMD simulation involves a good deal of brute force.* We conducted 100 μ s of HREMD simulations. This is one of the largest-scale enhanced sampling simulation in the literature. Other large-scale simulations only use 34 μ s of data [Chua *et al.*, *Proteins* **84**, 1134 (2016)] and 38 μ s of data [Chamachi, Chakrabarty *J. Phys. Chem. B* **120**, 7332 (2016)]. Our simulations are comparable to the work done in the group of D. E. Shaw [Pan *et al.*, *J. Chem. Theory Comput.* **12**, 1360 (2016)]. The rationale behind using brute force is that MD simulations are still comparably efficient in sampling, especially if the choice of the optimal bias is unclear.
- *Slow conformational changes occur in the bound state.* It is very likely that slowest conformational changes are those of the bound PMI-Mdm2 complex. However we do not have to sample these directly in the biased simulations. Instead of sampling a change between two bound conformations *A* and *B* directly, one can follow a thermodynamic cycle where one samples the transitions from *A* to loosely-bound and from loosely-bound to *B*. The replica exchange protocol that we use together with the biasing in the binding/unbinding direction allows the simulation to follow these alternative routes along the thermodynamic cycles spontaneously.
- *The system that we study is simple.* We pick the protein fragment ^{25–109}Mdm2 because it is relatively static. The fragment lacks the flexible lid formed by amino acids 1 to 24. The binding cleft is located on the surface of Mdm2, so there is no complicated and potentially dynamic binding channel that PMI has to penetrate. Furthermore we see almost no conformational changes in the Mdm2 fragment. The cleft motion that was observed in [Pantelopulos *et al.*, *Proteins* **83**, 1665 (2015)] plays no role in our simulations. The closed conformation is present in the apo-ensemble but with a minor population that is unlikely to have an influence on the binding kinetics. Unbound PMI changes conformation quickly.

In future research, we aim to develop more effective protocols for carrying out biased simulations, that involve TRAM. For instance we will explore to use HREMD-like simulations with longer contiguous trajectories and HREMD-like simulations that do not need to comply with the Monte-Carlo rule for swapping replicas but instead use TRAM’s built-in ability to deal with non-equilibrium data. There the choice of a good bias will be less important.

3. *In the abstract and elsewhere the authors mention that the binding comes from interchange between different conformations with different hydrophobic surfaces. Were all these conformations visited in the original 300 or 500 microsecond MD? Would there have been more conformations if the original MD was run 5000 microseconds, or 50,000 microseconds? How can the authors be sure about the sensitivity to this parameter? realize the authors compare the statistics from using 300 and 500 microsec runs, but if there are unvisited conformations even in 500 micro sec, this might not be sufficient. It would be nice to see a brief discussion on this point.*

The referee is completely correct that we can not be sure that all relevant conformations have been sampled and that the equilibrium probability might shift more and more to the bound state as new bound structures are discovered. However this criticism would apply to almost every simulation study except for very simple systems.

To address this concern, we have added the new Suppl. Fig. 7 where we show the fraction of metastable states seen in the trajectories depending on the total amount of simulation data analyzed. We find that the number of newly visited macro-states has almost reached 100%, once 60% of the simulation data is included in the analysis. This indicates that the majority of macro-states that are metastable on timescales of the microsecond and longer have been found in the simulation.

We have added a paragraph to the main text at the end of the section “Analysis of the full kinetic network” that describes this result and notes that it is still possible that further non-natively bound states exist:

“It is possible that the number of discovered non-natively bound structures, and their combined equilibrium probability, would continue to grow if the simulations would be extended. However, almost all metastable states found here are already visited in the first 60% of our simulation data (Suppl. Fig. 7) and the estimate for the binding free energy is converged (Fig. 2a). These indicators suggest that the non-natively bound structures with significant probabilities have been found.”

We want to stress that the application of TRAM (or TRAMMBAR) does not solve the sampling problem by itself. However these methods allow to connect states and to estimate transition rates in the forward and backward directions, and they can exploit the power of efficient sampling methods that have been developed over the years, such as metadynamics or MSM-driven adaptive sampling.

4. *What can the authors say about the unbinding mechanism instead of the binding mechanism, which (the former) is often of great pharmacological relevance? With how much confidence can they say this?*

In this work we studied the PMI-Mdm2 system under equilibrium conditions. In equilibrium the binding pathways and the unbinding pathways are identical due to detailed balance (microscopic reversibility). So there is no additional insight that can be gained from studying the unbinding mechanism. *In vivo* this is generally not true, because the steady-state concentrations of substrates and products may be out of equilibrium, thus violating detailed balance. However, the equilibrium kinetic model of a given protein-ligand interaction can be embedded into a larger kinetic model such as chemical master equations or particle-based reaction-diffusion models that involve irreversible steps to study the biological pathways (e. g. see [Boras *et al.*, *Front. Physiol.* **6**, 250 (2015)]). Secondly, the methods developed here can be employed to parametrization models for multivalent binders, which are often so strong that they can be practically considered irreversible. We have added a corresponding statement at the end of the Conclusions section.

5. *How much wall clock time did the original 500 microsecond MD take? Please mention this in the main text as I think this is an important aspect for practical applications.*

The simulation took approx. $115 \cdot 10^3$ GPU hours for the unbiased MD and approx. $42 \cdot 10^3$ GPU hours for the HREMD runs. We have added this information to the manuscript's Methods section.

The simulation speed was 105 ns/day for the unbiased simulations and 57.75 ns/day for biased simulations (single replica). The biased simulations are a bit slower because the bias potential had to be implemented on the CPU and not on the GPU.

In summary, I am happy with the paper but would like to see these few concerns alleviated before I can recommend publication.

We thank the referee for their supportive assessment.

Reviewers' Comments:

Reviewer #1 (Remarks to the Author):

The new draft is much improved. Congratulations to the authors. A few lingering comments:

- I recommend replacing reference 25 (Voelz et al.) with Bowman et al. JACS 2010, which captures 10 millisecond timescales with explicit solvent.
- In the paragraph that starts with path sampling, its not clear what "(B, C)" and other letters in parentheses refer to.
- I didn't have a problem with switching to implicit solvent, so much as switching between different variants of the amber protein force field for the simulations and free energy calculations. I suggest mentioning that this was done in the main text and providing a brief explanation as to why in the SI.

Reviewer #2 (Remarks to the Author):

The authors have addressed my concerns and I very gladly recommend publication of this most excellent manuscript.

Reviewer #1

The new draft is much improved. Congratulations to the authors.

We thank the referee for the supporting assessment.

A few lingering comments:

- *I recommend replacing reference 25 (Voelz et al.) with Bowman et al. JACS 2010, which captures 10 millisecond timescales with explicit solvent.*

We thank the referee for this suggestion. We have replaced the reference 25 (now reference 20 in the revised version of the manuscript) by [Bowman et al., Atomistic folding simulations of the five-helix bundle protein λ_{6-85} . *J. Am. Chem. Soc.* **133**, 664 (2011)], presuming that the referee meant 2011 and not 2010.

- *In the paragraph that starts with path sampling, its not clear what “(B, C)” and other letters in parentheses refer to.*

These letters refer to the tasks (A, B, C) introduced in the paragraph just above the one which starts with “path sampling”. To make it clearer what the letters refer to, we have added the word “task” to all references to tasks A–C in the revised manuscript.

- *I didn’t have a problem with switching to implicit solvent, so much as switching between different variants of the amber protein force field for the simulations and free energy calculations. I suggest mentioning that this was done in the main text and providing a brief explanation as to why in the SI.*

The motivation to switch to the ff14SB-olysc force field for the calculation of mutant energies was to use the most recently developed force field from the Simmerling group in the hope that it would give best agreement with experiment and that it would be the force field which would work optimally together with the generalized Born implicit solvent model (GB-Neck2, igb8) that was developed in the same group. Because we compute only potential energy *differences* $U_{\text{mut}}(x) - U_{\text{wt}}(x)$ from the ff14SB-olysc force field, energies that are not related to the removal of the side chain would cancel anyway and therefore and assumed that a change of force field would be valid. We had not realized that more recent research [Maffucci et al., *J. Chem. Theory Comput.*, **12**, 714 (2016)] has shown that the GB-Neck2 model works well together with the ff99SB-ILDN force field too.

Therefore we now have repeated the computation of differences in binding free energy upon alanine mutation of PMI with the ff99SB-ILDN force field, that was also used for our explicit solvent simulations. Results agree within statistical uncertainties and the qualitative picture and all conclusions remain the same. We have adapted main text figure 2e and the Supplementary table 2.

Differences between the $\Delta\Delta G$ values computed with the two force fields are shown in the figure below. No significant changes are observed. The largest difference between the force fields are found for the F3A mutation, where ff14SB-olysc yields $\Delta\Delta G = 8.7$ kcal/mol and ff99SB-ILDN yields $\Delta\Delta G = 5.3$ kcal/mol. The new value computed with ff99SB-ILDN is closer to the experimental value (5.5 kcal/mol, see also revised main text Fig 2e). Another change is the Y6A mutation where ff14SB-olysc yields $\Delta\Delta G = 1.5$ kcal/mol and ff99SB-ILDN yields $\Delta\Delta G = 0.1$ kcal/mol. Neither of the values agrees with the experimental value (3.1 kcal/mol).

Figure 1: $\Delta\Delta G$ upon mutation of PMI side chains to alanine computed with different force fields. Error bars show standard deviations.

Reviewer #2:

The authors have addressed my concerns and I very gladly recommend publication of this most excellent manuscript.

We thank the referee for their supportive assessment.